# Relationship between Blood Volume, Blood Lactate Quantity, and Lactate Concentrations during Exercise

**DOI:** 10.3390/metabo13050632

**Published:** 2023-05-06

**Authors:** Janis Schierbauer, Alina Wolf, Nadine B. Wachsmuth, Norbert Maassen, Walter F. J. Schmidt

**Affiliations:** 1Division of Exercise Physiology & Metabolism, University of Bayreuth, 95447 Bayreuth, Germany; 2Department of Sports Medicine/Sports Physiology, University of Bayreuth, 95447 Bayreuth, Germany; 3Institute of Sports Medicine, Hannover Medical School, 30625 Hannover, Germany

**Keywords:** lactate kinetics, hemoglobin concentration, hematocrit, plasma volume, erythrocyte volume, performance diagnostics

## Abstract

We wanted to determine the influence of total blood volume (BV) and blood lactate quantity on lactate concentrations during incremental exercise. Twenty-six healthy, nonsmoking, heterogeneously trained females (27.5 ± 5.9 ys) performed an incremental cardiopulmonary exercise test on a cycle ergometer during which maximum oxygen uptake ($\overset{\cdot }{V}$O_2max_), lactate concentrations ([La^−^]) and hemoglobin concentrations ([Hb]) were determined. Hemoglobin mass and blood volume (BV) were determined using an optimised carbon monoxide-rebreathing method. $\overset{\cdot }{V}$O_2max_ and maximum power (P_max_) ranged between 32 and 62 mL·min^−1^·kg^−1^ and 2.3 and 5.5 W·kg^−1^, respectively. BV ranged between 81 and 121 mL·kg^−1^ of lean body mass and decreased by 280 ± 115 mL (5.7%, *p* = 0.001) until P_max_. At P_max_, the [La^−^] was significantly correlated to the systemic lactate quantity (La^−^, r = 0.84, *p* < 0.0001) but also significantly negatively correlated to the BV (r = −0.44, *p* < 0.05). We calculated that the exercise-induced BV shifts significantly reduced the lactate transport capacity by 10.8% (*p* < 0.0001). Our results demonstrate that both the total BV and La^−^ have a major influence on the resulting [La^−^] during dynamic exercise. Moreover, the blood La^−^ transport capacity might be significantly reduced by the shift in plasma volume. We conclude, that the total BV might be another relevant factor in the interpretation of [La^−^] during a cardio-pulmonary exercise test.

## 1. Introduction

Lactate is now recognised as a major metabolic intermediate and signalling molecule that is used for oxidative energy supply and is transported between different cells, tissues, and organs by means of transport proteins [1,2,3,4,5,6,7]. Lactate is widely accepted as a diagnostical marker and lactate kinetics or lactate concentrations ([La^−^] are commonly used in the field of sports medicine for the assessment of endurance performance, e.g., during an incremental cardiopulmonary exercise (CPX) test [8]. Moreover, exercise prescriptions based on [La^−^] allow for precise and predictable regulation of acute metabolic and cardiorespiratory responses during dynamic exercise, as is reflected in the previously postulated lactate turn point model [8,9].

However, [La^−^] must always be considered in light of the prevailing production and elimination rates and numerous studies have shown that there are several factors that can significantly influence these rates leading to highly variable results. These include, for example, the applied protocol or workload characteristics [10,11,12] of a CPX test, the previous diet in relation to muscle glycogen [13,14,15,16], muscle fibre-type composition [17,18,19], the source of blood sampling [12,20,21], or cerebral lactate uptake [22].

Another important factor that has not yet been considered in this context may be the total blood volume (BV). This is surprising since the BV not only serves as a distribution medium but also transports lactate to other cells and organs for the oxidative energy supply or gluconeogenetic metabolism.

It can be assumed that the concentration of any substance dissolved in a medium is dependent on both the size of the medium and the amount of substance dissolved in the medium. Usually, [La^−^] is measured throughout an exercise test regardless of the medium, in this case, total BV. It is well known, however, that total BV not only differs greatly between individuals, e.g., due to training-induced volume expansion or a genetic predisposition [23,24,25,26] but also decreases by up to 10% during dynamic exercise mainly as a result of plasma volume (PV) shifts [27,28]. With respect to the latter, Davies et al. previously concluded that [La^−^] above the lactate threshold should be corrected for this decrease in PV [29]. Moreover, since significantly more lactate is transported in the plasma than in the erythrocytes, especially during intensive exercise [30], PV losses might also have a considerably negative impact on the lactate transport capacity. Consequently, the total BV may have an impact on both inter- and intra-individual comparisons of [La^−^]. Therefore, we hypothesise that the larger (or the smaller) the total BV, the lower (or the higher) the [La^−^] tends to be in the course of an incremental ergometer test. In addition, if the BV is known, it is also possible to calculate the absolute lactate quantity (La^−^) and thus, its impact on [La^−^]. To the best of our knowledge, this has not yet been done. Therefore, the aim of this study was to determine the total BV and La^−^ during an incremental CPX test on a cycle ergometer in healthy volunteers with heterogenous endurance capacities and quantify their influence on the measured [La^−^] during dynamic exercise.

## 2. Materials and Methods

### 2.1. Participants

This was a secondary outcome analysis of a previously published descriptive cross-sectional study [31] that reports preliminary observations. Twenty-six healthy, nonsmoking females with heterogenous endurance capacity and no history of cardiac disease were included in the study (see Table 1 for participant characteristics). The participants provided written consent after being informed of the study design, the associated risks, and their right to withdraw at any time. The study was conducted in conformity with the declaration of Helsinki and Good Clinical Practice and the study protocol was approved by the ethics committee of the University of Bayreuth in Germany (O 1305/1–GB).

### 2.2. Study Design

After anthropometric measurements including analysis of body composition using a bioelectrical impedance analysis were conducted, a cubital venous blood sample was drawn for a full blood count as well as ferritin concentrations to exclude any iron deficiencies. The participants then performed an incremental CPX test on a cycle ergometer to determine the maximum power (P_max_) and maximum oxygen uptake ($\overset{\cdot }{V}$O_2max_). During this test, the hemoglobin concentration ([Hb]) for the calculation of BV and capillary lactate concentrations ([La^−^]) were determined. The hemoglobin mass (Hbmass) was measured twice on consecutive days and within 7 days after the ergometer test using a CO-rebreathing method. The BV at rest and during exercise was calculated subsequently based on the Hbmass and [Hb].

### 2.3. Anthropometry and Blood Sampling

Prior to the exercise test, lean body mass (LBM) and fat mass were measured twice consecutively using a bioelectrical impedance analyser (InBody 720, InBody Co., Seoul, Republic of Korea). Cubital venous blood samples (8 mL) were drawn after the participants rested for 15 min in an upright seated position. Heparinised blood samples were analysed using a fully automated hematology system (Sysmex XN 1000-1-A, Sysmex, Norderstedt, Germany) for red blood cells including hemoglobin concentration ([Hb]) and hematocrit (Hct). The serum ferritin and C-reactive protein (CRP) concentrations were determined by enzyme immunoassays [ferritin: LKFE1, CRP: highly sensitive—LKCRP1; ELISA & Immulite 1000 (Siemens Healthcare Diagnostics GmbH, Erlangen, Germany)].

### 2.4. Cardio-Pulmonary Exercise Test and Lactate Analysis

P_max_ was determined using an incremental protocol on a cycle ergometer (Excalibur Sport, Lode, Groningen, The Netherlands). After a 3-min warm-up phase of 50 W, the mechanical power was increased by 50 Watts every 3 min (stepwise by 17, 17, and 16 Watts per minute) until subject exhaustion was reached. The VO_2_ was determined via breath-by-breath technology (Metalyzer 3B, Cortex, Leipzig, Germany), and the $\overset{\cdot }{V}$O_2max_) was calculated as the highest 30 s interval before exhaustion. Capillary blood samples were taken from a hyperemised earlobe before exercise, every 3 min during exercise, and immediately at exhaustion to determine the [Hb] using a calibrated photometric analysis (HemoCue 201, Hemocue AB, Ängelholm, Sweden). At the same time points, capillary blood samples (20 µL) were taken from the other earlobe to measure the [La^−^] using an enzymatic-amperometric approach (Biosen S-Line, EKF-Diagnostic, Barleben, Germany). Further blood samples for the determination of [La^−^] were taken 1-, 3-, 5- and 7-min post-exercise. The maximum lactate concentration was defined as [La^−^]_max_. The absolute lactate quantity (La^−^) in mmol at the respective intensities was calculated as the product of [La^−^] and BV and indexed for lean body mass. The [La^−^] and La^−^ at 60% of P_max_ (P_60%_) and at P_max_ were defined as [La^−^]_60%_ and La^−^_60%_ and [La^−^]_end_ and La^−^_end_, respectively. The lactate transport capacity in the erythrocyte and in the plasma volume (PV) was calculated based on the blood volume at P_max_ (BV_end_), the Hct_end,_ and the corresponding [La^−^]. Additionally, the [La^−^] and La^−^ in the PV and the erythrocyte volume (ECV) at P_max_ were separately estimated assuming a [La^−^] ratio of 1:0.3 between the PV and the ECV [30].

### 2.5. Determination of Hemoglobin Mass and Blood Volume

The Hbmass, BV, PV, and ECV were determined using a carbon monoxide (CO)-rebreathing procedure according to methods described in previous investigations [32,33,34]. In brief, an individual dose of CO (0.8–0.9 mL·kg^−1^, CO 3.7, Linde AG, Unterschleißheim, Germany) was administered and rebreathed along with 3 L of pure medical oxygen (Med. O_2_ UN 1072, Rießner-Gase GmbH, Lichtenfels, Germany) for 2 min. Capillary blood samples were taken before and 6 and 8 min post administration of the CO dose. The blood samples were measured for the determination of %HbCO using an OSM III hemoximeter (Radiometer, Copenhagen, Denmark). The Hbmass was calculated based on the mean change in %HbCO before and after the CO was rebreathed. As part of the equation to calculate changes in BV during the exercise period, the capillary [Hb] was converted to the venous conditions [35,36]. The Hct_end_ was calculated as the quotient of [Hb]_end_ and the mean corpuscular hemoglobin concentration (MCHC) at rest [37]. The BV was calculated according to the following formula where 0.91 = cell factor at sea level [38]:$$BV\left(mL\right)=Hbmass\left(g\right)\times 100÷[Hb]{\left(g\cdot dL\right)}^{-1}÷0.91$$

The BV at rest and at P_60%_ were defined as BV_rest_ and BV_60%_, respectively. The BV at maximum power was defined as BV_end_. For the calculation of the BV_60%_, the [Hb], which was determined at rest and every 3 min during exercise, was interpolated for the respective exercise intensity, if necessary. Hbmass was measured twice on consecutive days and within 7 days after the ergometer test with a possible first test at least 2 h after the CPX test when the plasma volumes had returned to pre-exercise values [39]. Since the Hbmass does not change over short periods [40], the temporary offset determination of the [Hb] for the calculation of the BV is possible without compromising accuracy. For a detailed description and the accuracy of the methods see [32,33,34]. The typical error for Hbmass in our laboratory is 1.5%, which is comparable to previous investigations [35,41], while the typical error for BV is 2.5%.

### 2.6. Statistical Analyses

The data are presented as means and standard deviations. Statistical analysis was conducted using GraphPad Prism Version 8.0.2 (GraphPad Software, Inc., San Diego, CA, USA). Testing for normality was performed using the Shapiro-Wilk test. Pearson correlation coefficients or nonparametric Spearman correlations were computed to prove any relationship between the two variables. A paired t-test was computed to calculate the differences in La- with and without the BV shifts. Multiple linear regression was performed to predict the value of one dependent variable (e.g., [La^−^]) based on two independent variables (e.g., BV and La^−^). The level of significance was set to *p* ≤ 0.05.

## 3. Results

$\overset{\cdot }{V}$O_2max_ ranged between 32 and 62 mL·kg^−1^·min^−1^. P_max_ ranged between 2.3 and 5.5 W·kg^−1^. Maximum heart rate ranged between 159–201 beats per minute, and the respiratory exchange ratio at P_max_ ranged between 1.13 and 1.34. The Hbmass ranged between 7.8 and 12.7 g·kg^−1^. The BV_rest_ decreased by 280 ± 115 mL (5.7%, *p* = 0.001) until P_max_ leading to an increase in [Hb] by 0.8 ± 0.3 g·dL^−1^ (*p* < 0.001). Data on the BV, La^−^ and [La^−^] at submaximal and maximal power can be found in Table 2.

As depicted in Figure 1A, [La^−^]_end_ was significantly correlated to La^−^_end_ (y = 10.2x + 0.2378, r = 0.84, *p* < 0.0001). However, we found no correlation between the La^−^_end_ and BV_end_ (Figure 1B). In contrast, the [La^−^]_end_ was significantly and negatively correlated to BV_end_ (y = −0.104x + 21.34, r = −0.44, *p* < 0.05, Figure 1C). Similar results were found between [La^−^]_60%_ and La^−^_60%_ (y = 20.2x + 0.139, r = 0.92, *p* < 0.0001), however, the correlation between [La^−^]_60%_ and BV_60%_ was not significant. As a result of the exercise-induced BV shifts, the [La^−^]_60%_ and [La^−^]_end_ were 0.12 ± 0.09 and 0.66 ± 0.29 mmol·L^−1^ higher, respectively, when compared to the theoretical situation had the plasma volume and La^−^ been unchanged.

The lactate quantity (La^−^_end_) in the total BV at maximum power was 51.8 ± 12.2 mmol. It was 10.8% lower when compared to the calculated La^−^_end_ with unchanged plasma volume and unchanged lactate concentration (56.3 ± 13.6 mmol, *p* < 0.0001, Figure 2). Multiple linear regression revealed that, when indexed for lean body mass, both BV_end_ (β = −0.1244, *p* < 0.0001) and La^−^_end_ (β = 10.78, *p* < 0.0001) significantly influence [La^−^]_end_. For the submaximal exercise intensity, significant results were only observed for La^−^_60%_ (β = 19.71, *p* < 0.0001).

## 4. Discussion

This study aimed to calculate the influence of total BV and absolute lactate quantity on the measured [La^−^] during an incremental CPX test on a cycle ergometer in healthy volunteers with heterogenous endurance capacities. Our findings confirm the theoretical assumption that both the blood volume and the lactate quantity have an impact on the resulting [La^−^]. In addition, the exercise-induced BV shifts led to a significant 10.8% reduction in the lactate transport capacity.

The measured [La^−^] in the blood is generally the result of lactate production (e.g., within the muscle cells) and lactate elimination (e.g., the diffusion of lactate into the PV and its distribution to other cells and organs for lactate clearance). The latter also includes an individual’s total BV as a distribution space and the exercise-induced BV shifts. Previous investigations have already dealt in detail with various factors influencing lactate production and elimination rates [12,13,19,42,43], however, to the best of our knowledge, this is the first study that will focus exclusively on the potential role of the BV.

**The role of blood volume as distribution space:** In theory, if the exchange of lactate between cells remains constant, then a higher BV would consequently lead to lower [La^−^] and vice versa. On the other hand, for a given [La^−^], a higher La^−^ would also be associated with a higher BV and vice versa. Our data demonstrate that both assumptions are equally true, however, there are quantitative differences between the two mechanisms.

As demonstrated in the multiple linear regression analysis, both the BV_end_ (β = −0.1244, *p* < 0.0001) and La^−^_end_ (β = 10.78, *p* < 0.0001) significantly influence the [La^−^]_end_. With regard to the correlation between the La^−^_end_ and BV_end_ (Figure 1B), however, no significance was found. This finding might be explained by a plateau in net lactate release at maximum exercise indicating a disturbance in lactate exchangeability [44,45,46]. In contrast, the larger lactate quantity is not sufficient to align the [La^−^] with different BV, which is confirmed by the significant negative correlation between [La^−^]_end_ and BV_end_ (Figure 1C). Therefore, a higher BV might have two opposing effects: First, it leads to a greater diffusion gradient allowing for more lactate to diffuse out of the muscle cell; second, it also increases the distribution space, which is reflected in the lower [La^−^] (Figure 1C). However, this does not consider simultaneous lactate extraction and net release within the same muscle fibre types of the same exercising muscle groups. In endurance-trained individuals, where this mechanism can be substantially enhanced, this would lead to a reduced systemic La^−^ that further reduces the measured [La^−^] in addition to their larger BV.

**Contribution of ECV and PV to lactate transport:** [La^−^] are normally measured in the whole blood. However, erythrocytes show a lower [La^−^] when compared to plasma [La^−^]. This difference in [La^−^] between plasma and erythrocytes was found to be 1:0.5 under resting conditions but is augmented by strenuous exercise, i.e., 1:0.2 [30]. Accordingly, both volumes are contributing to the distribution of lactate at submaximal exercise intensities. With increasing exercise intensity, however, the aforementioned ratio changes substantially making the plasma volume almost exclusively responsible for the lactate transport at maximal exercise [47]. The significant reduction in plasma volume further decreases blood lactate transport capacity. We have calculated that even when a more cautious ratio of 1:0.3 is used, the lactate transport capacity in the BV at maximum exertion was still significantly reduced by 10.8% exclusively as a result of the PV reduction (Figure 2). It was previously argued that the erythrocyte membrane provides a barrier to the flux of lactate between PV and ECV during rapidly changing blood lactate levels [47]. Thus, in addition to the well-known thermoregulatory and cardiovascular disadvantages of a reduced plasma volume [48,49], our findings imply that it may also have a detrimental effect on lactate transport capacity.

**Confounding factors:** Two of our subjects with nearly identical La^−^_end_ (1.02 vs. 1.04 mmol·kg^−1^) showed distinct differences in BV_end_ when indexed for lean body mass (94.2 and 116.5 mL·kg^−1^), thus leading to different [La^−^]_end_ (10.9 and 8.9 mmol·L^−1^). However, these two participants also differed in their $\overset{\cdot }{V}$O_2max_ (36.2 and 59.4 mL·min^−1^·kg^−1^) and P_max_ (3.2 and 5.0 W·kg^−1^), respectively. Since their La^−^_end_ were identical, the differences in [La^−^]_end_ in these individuals can most likely be attributed to different blood volumes. Therefore, in order to understand the relationship between [La^−^] and La^−^, a more holistic approach is required. As mentioned before, the maximum systemic lactate concentration, e.g., when measured in the capillary blood, is always the result of lactate production, exchange, and utilisation [7,50] and the absolute amount of lactate in the blood depends on the interaction between the aforementioned factors.

With regard to the role of total BV, both intra- and inter-individual factors have to be considered. For instance, it was demonstrated particularly in endurance-trained individuals that the lower blood [La^−^] is usually the combined result of a training-induced decrease in the overall release of lactate from tissues to blood as well as an increase in clearance from plasma during exercise [51] equalling a lower absolute La^−^ in the blood. This is most likely due to an increase in mitochondrial monocarboxylate transporters and enzymatic lactate dehydrogenase activity which in turn improves the oxidative capacity of the muscle cells [52]. In addition to their large BV, these adaptations most likely lead to chronically lower [La^−^] during dynamic exercise.

Similar conditions can also be observed on the inter-individual level, e.g., in trained athletes from different disciplines. While athletes participating in sports requiring explosive muscular power, e.g., sprinting exercise, usually possess a larger quantity of high glycolytic fibre types, elite endurance athletes are characterised by a much higher percentage of oxidative fibres. Additionally, sprinters are characterised by a lower BV when compared to endurance-trained athletes [53]. Moreover, their PV losses are typically larger than during endurance exercise reducing the most effective distribution medium for La^−^ even further [54]. This would suggest that the usually higher [La^−^] found in sprinters at a higher power do not necessarily indicate a higher absolute blood La^−^ when compared to endurance-trained athletes. The same would eventually apply for inter-individual comparisons between untrained individuals with different BV, e.g., due to genetic predisposition [23].

**Practical Implications:** Although BV is not routinely determined in most exercise labs, this study highlights its general relevance in the context of lactate diagnostics and the interpretation of results. In terms of blood lactate diagnostics in exercise testing and training it should be noted, that the PV can increase by up to 15% as a result of systematic endurance training [25]; while the ECV can increase by up to 6% [55,56]. Notably, the PV significantly increases after just very short time periods (hours to days), thus increasing the potential distribution volume rapidly long before metabolic adaptations take place [57]. Therefore, the increases in BV should also be considered in the interpretation of changes in lactate kinetics, especially when (long-term) training interventions are conducted. This is independent of the fact that the percentage differences in La^−^ as a result of the exercise-induced PV shifts that we found in this study likely yield no practical importance for interpreting lactate kinetics. Moreover, our results underline the importance of ensuring an adequate hydration state before and during training and competition. Here, PV is important for supporting the homeostasis of cardiovascular and thermoregulatory systems [58]. For instance, it has been shown that an isotonic reduction in PV in turn leads to a reduction in sweat rate [59]. These findings may be of even greater importance when exercising in the heat [60] or monitoring exercise intensity using non-invasive biomarkers such as forehead sweat lactate secretion rate [61]. Lastly, it would be of great interest as to whether a threshold determination based on absolute systemic La^−^ indexed for lean body mass rather than [La^−^] could be a viable option in the context of lactate performance diagnostics.

**Limitations:** There are several limitations to this study. First, when interpreting our data, it is important to consider that our assumptions have been made on selected exercise intensities during an incremental CPX test, i.e., P_60%_ and P_max_. Second, it may well be that the dependencies between the BV, La^−^ and [La^−^] diverge with regard to different exercise modalities, e.g., during a Wingate test, high-intensity interval exercise, or a continuous moderate exercise, due to different lactate fluxes from working muscles to the bloodstream, different ratios between PV and erythrocytes, or different fluxes of lactate to consuming tissues [43]. Third, our study population consisted exclusively of heterogeneously trained female participants. Although we would hypothesise that the same mechanisms also apply to men, especially since it was demonstrated that their BV shifts are typically much larger [27,28,62], we have not controlled for the menstrual cycle which has been shown to have an influence on water retention, and thus plasma volume [63]. Moreover, inter-individual differences in training status may also have affected the rate of lactate production and clearance during exercise thus affecting our results [64].

## 5. Conclusions

In this study, we evaluated the influence of blood volume and absolute systemic lactate quantity on the lactate concentrations during an incremental CPX test in healthy individuals. Our findings demonstrate that a higher BV was associated with a lower [La^−^] at maximum exercise. Since the [La^−^] between erythrocyte and plasma substantially differ during intensive exercise, acute plasma volume changes also have a substantial influence on the lactate transport capacity in the total blood volume. In addition to its influence on cardiovascular and thermoregulatory stability, our data indicate that another important property can be attributed to PV.

## Figures and Tables

**Figure 1 metabolites-13-00632-f001:**
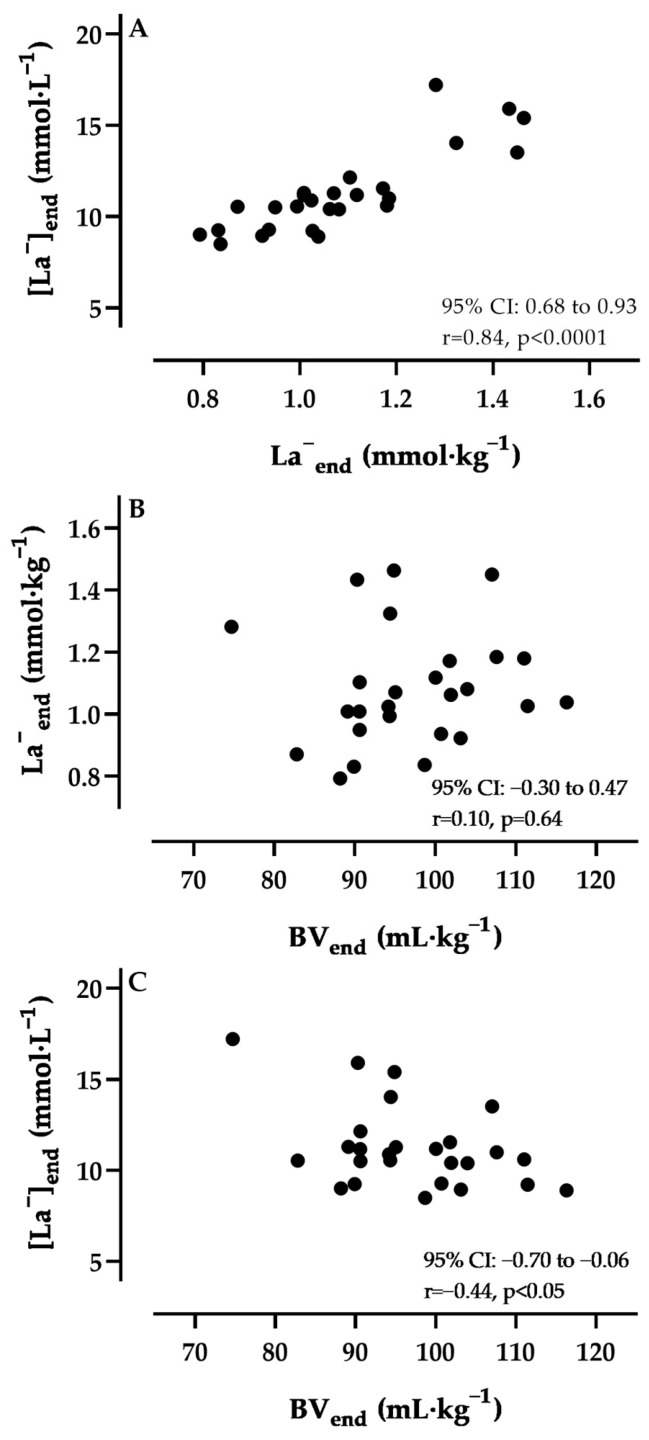
Correlation coefficients for the lactate concentrations ([La^−^]_end_) and lactate quantity (La^−^_end_) (**A**), La^−^_end_ and blood volume (BV_end_) (**B**), and [La^−^]_end_ and BV_end_ (**C**). Data were indexed for lean body mass.

**Figure 2 metabolites-13-00632-f002:**
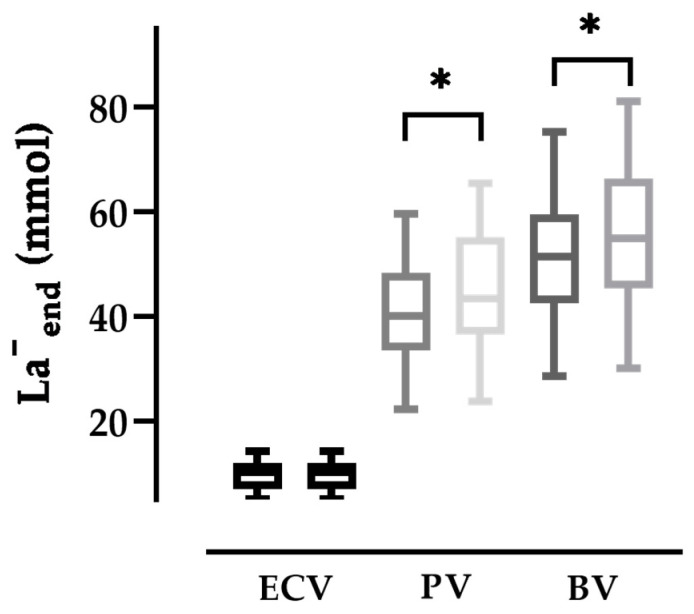
Blood lactate quantity at maximum power (La^−^_end_) in the erythrocyte volume (ECV), plasma volume (PV), and total blood volume (BV) with (left) and without (right) blood volume shifts (*p* < 0.0001). La^−^_end_ was initially calculated as the product of BV_end_ and [La^−^]_end_ (* indicates *p* < 0.0001). The displayed values for La^−^_end_ were estimated assuming a [La^−^] ratio of 1:0.3 between the plasma and erythrocyte [30].

**Table 1 metabolites-13-00632-t001:** Participant characteristics (n = 26).

	Mean ± SD	Min	Max	95% CI
Age (y)	27.5 ± 5.9	19	40	25.1–29.9
Height (cm)	167.7 ± 6.5	154	180	165–170
Body mass (kg)	60.1 ± 7.0	47.5	73.5	58.1–63.9
Body mass index (kg·m^−2^)	21.6 ± 1.6	18.6	25.1	20.9–22.3
Lean body mass (kg)	47.4 ± 5.9	35.9	56.9	44.9–49.9
Fat mass (%)	22.2 ± 5.6	9.4	35.0	19.8–24.6
Ferritin (μg·L^−1^)	44 ± 24	16	105	34.2–54.0
C-reactive protein (mg·dL^−1^)	1.37 ± 1.26	0.3	4.9	1.43–2.52
$\overset{\cdot }{V}$O_2max_ (mL·min^−1^·kg^−1^)	49.0 ± 8.1	31.8	61.7	45.7–52.3
P_max_ (W·kg^−1^)	4.2 ± 0.8	117	334	236–282
Hbmass (g·kg^−1^)	9.8 ± 1.2	7.8	12.7	9.3–10.3

The data are presented as the mean values ± standard deviations. Min = minimum, Max = maximum, CI = confidence interval.

**Table 2 metabolites-13-00632-t002:** Blood lactate and blood volume at rest and during the cardio-pulmonary exercise test.

	Mean ± SD	Min	Max	95% CI
[La^−^]_60%_ (mmol·L^−1^)	2.5 ± 0.9	1.1	4.5	2.1–2.9
La^−^_60%_ (mmol)	11.4 ± 4.2	5.6	20.8	9.6–13.1
La^−^_60%_ (mmol·kg^−1^ LBM)	0.11 ± 0.04	0.06	0.22	0.09–0.14
[La^−^]_end_ (mmol·L^−1^)	11.3 ± 2.2	8.5	17.2	10.3–12.1
La^−^_end_ (mmol)	52 ± 12.2	28.6	75.3	46.7–56.8
La^−^_end_ (mmol·kg^−1^ LBM)	1.08 ± 0.19	0.79	1.46	1.01–1.16
[La^−^]_max_ (mmol·L^−1^)	12.1 ± 2.4	8.5	18.4	11.2–13.1
BV_rest_ (mL)	4889 ± 836	3306	6461	4551–5227
BV_rest_ (mL·kg^−1^ LBM)	102.0 ± 9.9	81	121	99–107
BV_end_ (mL)	4609 ± 799	3063	6298	4286–4932
BV_end_ (mL·kg^−1^ LBM)	97.0 ± 9.5	75	116	92–100
[Hb]_rest_ (g·dL^−1^)	13.4 ± 0.75	11.5	15.3	13.1–13.7
[Hb]_end_ (g·dL^−1^)	14.2 ± 0.78	12.2	16.0	13.9–14.5

SD = standard deviation, Min = minimum, Max = maximum, CI = confidence interval of the mean, [La^−^]_60%_ = lactate concentration at P_60%_, [La^−^]_end_ = lactate concentration at maximum exercise, La^−^_60%_ = lactate quantity at P_60%_, La^−^_end_ = lactate quantity at maximum exercise, [La^−^]_max_ = maximum lactate concentration, BV_rest_ = blood volume at rest prior to exercise, BV_end_ = blood volume at maximum exercise, LBM = lean body mass, [Hb] = hemoglobin concentration.

## Data Availability

Data will be made available upon reasonable request by the corresponding author. Data is not publicly available due to privacy or ethical restrictions.

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
