# Peer review of "Relationship between Blood Volume, Blood Lactate Quantity, and Lactate Concentrations during Exercise"

_metabolites, 2023, doi:10.3390/metabo13050632_

Round 1

Reviewer 1 Report

At first glance, the work seems to be correct. Authors described the relationships between the lactic acid blood concentration ([La-]), blood lactate quantity (La-) and total blood volume (BV) in females studied during a cycle ergometer test. They have found that the total BV and La- influenced the [La-].

But the La- wasn’t measured by direct method. The value of this parameter was calculated on the basis of [La-] and BV values. So their values are necessarily related.

Next observation concerned the effect of the exercise-induced shift in plasma volume on the blood lactic acid (La) transport capacity. I am not convinced of the truth of this statement. The La is well soluble in water. The fluid that leaves the vascular bed during intense exercise and penetrates into the intercellular space outside the vascular bed contains dissolved La, as do sweat, urine, and saliva. The process of exchange of components between intravascular and extravascular fluids is still ongoing. Moreover, a reduced amount of intravascular fluid can carry larger amounts of La, because it is highly soluble, so there can be no talk of a reduction in the transport capacity of blood.

Authors concluded that “for the holistic interpretation of [La-] during exercise, the BV should be incorporated as an additional influencing factor”.

In my opinion, it is a very general statement. It is obvious that differences in the amount of solvent affect the concentration of the solution. However, the authors did not take into account other important factors, such as the loss of La and water in sweat, urine, saliva, etc. The methods used to measure the lean body mass, fat mass, BV, La- and hemoglobin mass are indirect methods and are subject to an error of at least 5%. Moreover, the group of participants included heterogeneously trained individuals. It is known that the ability to La production, and anaerobic threshold depend on the level of performance. The level of training also affects the rate of lactate utilization during and after exercise.  The phase of the menstrual cycle was also not taken into account, and yet it affects the retention of water in the body. In conclusion, the failure to take into account the many factors affecting the amount of water in the body and the use of estimated rather than directly measured parameters undermines the significance of the results of this work.

If, despite the doubts presented, this work will be accepted for publication, the following errors should be corrected:

In Table 2:

95% CI of BVend (mL·kg-1 LBM) is 32 – 100 – it is mistake, it should much more, maybe 92-100?

In lines 218 – 220 – absolutely, there is no trend here. The value of p = 0.64, and not < 0.10. Please remove this statement.

Author Response

Dear Reviewer and esteemed colleague,

first of all, we would like to thank you very much for your time and effort in reviewing this manuscript. We have found the comments to be very helpful in terms of improving the overall quality of the paper. We have taken all comments into account and revised the manuscript accordingly. Enclosed, you may find our reply document.

Kind regards

Janis Schierbauer et al.

Reviewer 2 Report

Thank you for the opportunity to review this article, the authors presented very interesting research.

·      It is written correctly. Gives highlights from each section of the paper. 

·      The study group is too small, authors should add in the title “preliminary observations” or “pilot study”.

·      Introduction 

·      It is written correctly. However, it should be developed better.

·      Method

·      Identify a clear study design.

·      I would suggest to the authors to clarify the sample recruitment procedures.

·      How was the power of the drafted sample calculated? (G-Power)?

·      In my opinion, including a table showing the paired t-test analysis would make the results obtained clearer.

·      Authors should complete the limitations of this study (insert if the sample size is small).

·      References - please refer to other publications.

Author Response

(The authors gave the same response as above.)

Reviewer 3 Report

The topic of this article is interesting, but not a novelty, the authors presenting the results of a clinical study performed on twenty-six healthy females subjected to exercise test, on cycle ergometer, aiming to investigate the influence of blood volume on the lactate concentration.

The present style of manuscript cannot be accepted for publication due to several reasons described below. I recommend the authors the major revision to blush the manuscript up.

The title of the article should be reconsidered to make it more relevant.

The summary must be redone, with a brief mention of the protocol of the study, the synthetic results and, in particular, the elaboration of clear and relevant conclusions. The methods used for the assessing of the biological parameters, as well as their values, and the statistical significance, should be removed from the abstract.

The introduction section should be more complete, providing supplementary background in the field.

The results obtained should be compared with those achieved by other researchers and discussions should be significantly detailed.

(see:

·   Lee S, Choi Y et al. Physiological significance of elevated levels of lactate by exercise training in the brain and body. J Biosci Bioeng 2023; 135(3): 167-175.

·   Yamagata T et al. Relationship between sweat lactate secretion rate and blood lactate concentration during exercise near the lactate threshold. Gazzetta Medica Italiana - Archivio per le Scienze Mediche 2022; 181(11): 833-40

·   Durand R et al. Modelling of blood lactate time-courses during exercise and/or the subsequent recovery: limitations and few perspectives. Front Physiol 2021; 12: 702252.

·   Siebenmann C et al. Cerebral lactate uptake during exercise is driven by the increased arterial lactate concentration. J Appl Physiol 2021; 131(6): 1824-1830.

and others)

In the discussion section, the authors need to develop argumentation in depth based on the current understanding and the findings of the results obtained, presenting the potential, the weakness and future research direction, among others. Authors should try to explain the theoretical implication as well as the translational application of their research.

Some other aspects were found in this manuscript:

- all abbreviations should be expanded in the first appearance and should not be repeated in order to decongest the text and facilitate the understanding of the information transmitted;

- in the text, the terms coded [La-], (La-), La-, [La-]end and others were not clearly defined, thus create confusion;

- the information regarding the date of obtaining the Ethical Certificate is missing;

- different fonts were used in the text and in the figures;

- the references should be upgraded;

- at the references the authors should provide the DOI of the all the articles

- a schematic representation of the study would be appreciated;

- spelling check of the text is mandatory;

- English including grammar, style and syntax, should be improved through the professional help from English Editing Company for Scientific Writings.

Author Response

(The authors gave the same response as above.)

Round 2

Reviewer 1 Report

The indicated errors have been eliminated, which does not change my skeptical attitude towards the whole work and its overtones.

In my opinion, the results and conclusions of this work are of no practical importance.

The authors used elaborate measurement and estimation methods, including the need for study participants to inhale carbon monoxide, just to show the obvious relationship between the amount of lactate produced, blood volume, and blood lactate concentration. Moreover, this complex analysis allows for possible correction of the blood lactate results obtained in the range of 10%. A 10% difference in blood lactic acid measurements is of no practical importance. Such information should be included in the Practical Implications chapter.

Author Response

Dear Reviewer,

once again, thank you very much for your time and effort in reviewing this manuscript. We have updated the mentioned section in the manuscript.

Kind regards

Janis Schierbauer et al.

Reviewer 2 Report

the manuscript is much clearer after the revisions, advice to acceptance

Author Response

Dear Reviewer,

thank you very much for your time and effort.

Kind regards

Schierbauer et al.

Reviewer 3 Report

The authors have significantly revised the manuscript addressing the concern raised. I consider it could be accepted for publication in this journal.

Author Response

(The authors gave the same response as above.)
